# Rates of subsequent surgeries after meniscus repair with and without concurrent anterior cruciate ligament reconstruction

Joseph B. Kahan[1], Patrick Burroughs[2], Logan Petit[1], Christopher A. Schneble[1], Peter Joo[1], Jay Moran[1], Maxwell Modrak[1], William Mclaughlin[1], Adam Nasreddine[1], Jonathan N. Grauer[1], Michael J. Medvecky[1]*

1 Department of Orthopaedics and Rehabilitation, Yale School of Medicine, New Haven, CT, United States of America, 2 Yale School of Medicine, New Haven, CT, United States of America

* Michael.Medvecky@yale.edu

**Data Availability Statement:** The data underlying the results presented in the study are available through the third party vendor PearlDiver Inc. (URL: https://pearldiverinc.com/). The data may be

## Abstract

### Objectives

The purpose of this study was to compare the rates of secondary knee surgery for patients undergoing meniscus repair with or without concurrent anterior cruciate ligament reconstruction (ACLr).

### Methods

Utilizing a large national database, patients with meniscal repair with or without concurrent arthroscopic ACLr were identified. The two cohorts were then queried for secondary surgical procedures of the knee within the following 2 years. Frequency, age distribution, rates of secondary surgery, and type of secondary procedures performed were compared.

### Results

In total, 1,585 patients were identified: meniscus repair with ACLr was performed for 1,006 (63.5%) and isolated meniscal repair was performed for 579 (36.5%). Minimum of two year follow up was present for 487 (30.7% of the overall study population).

Secondary surgery rates were not significantly different between meniscus repair with concurrent ACLr and isolated meniscus repairs with an overall mean follow up of 13 years (1.5–24 years) (10.6% vs. 13.6%, p = 0.126). For the 2 year follow up cohort, secondary surgery rates were not significantly different (19.3% vs. 25.6%, p = 0.1098). There were no differences in survivorship patterns between the two procedures, both in the larger cohort (p = 0.2016), and the cohort with minimum 2-year follow-up (p = 0.0586).

### Conclusion

The current study assessed secondary surgery rates in patients undergoing meniscus repair with or without concurrent ACLr in a large patient database. Based on this data, no significant difference in rates of secondary knee surgery was identified.

purchased via the third party vendor and queried through the provided software using CPT and ICD coding. The authors did not have special permission or privileges outside of those granted via payment to the vendor.

**Funding:** The authors received no specific funding for this work.

**Competing interests:** I have read the journal's policy and the authors of this manuscript have the following competing interests: Medvecky guest speaker/honoraria for Smith & Nephew. This does not alter our adherence to PLOS ONE policies on sharing data and materials.

## Introduction

Meniscal tears are the most commonly treated knee injury in the United States, with an incidence between 60 and 70 per 100,000, and approximately 850,000 meniscal procedures performed annually [1–4]. The majority of such procedures are meniscal debridement, however in select cases meniscal repair can be considered [5–8]. The durability of such repair procedures has been quoted to be between 84–91% at two years not requiring subsequent surgery [9,10]. In prior studies, a subset of meniscus repairs have been found to be incompletely healed but the patients can remain asymptomatic [11,12]. If there is not biologic healing and further intervention is needed, a meniscectomy is most frequently performed. However, revision meniscus repair, meniscus transplantation and knee arthroplasty can be viable alternatives for the failed meniscus repair, depending upon the status of the articular cartilage [13,14].

Meniscal injury occurs in approximately 50% of patients with an acute anterior cruciate ligament (ACL) injury [15,16]. Meniscal repair performed concurrent with anterior cruciate ligament reconstruction (ACLr) has received specific attention. Some studies have found that healing rates of meniscus repairs with concurrent ACLr are higher than isolated meniscus repair [10,17,18]. This has been postulated to be due to the rich biologic environment created during reconstruction. In their prospective matched cohort study of approximately 1250 patients in each cohort, Wasserstein et al. found higher rates of secondary knee surgery in patients undergoing isolated meniscus repair (16.7%), compared to a combined meniscus repair and ACLr (9.7%) at 2 year follow up [10].

However, other studies of clinical failure rates of isolated repairs and those performed concurrently with ACLr have found them to fare similarly over time [9,19,20]. With minimum follow up of five years, Bogunovic et al. found a failure rate of 12% with isolated repairs and a 18% failure rate with combined ACL reconstructions in a total of 75 patients, although this difference was not statistically significant [19].

Overall, the literature is inconsistent about whether meniscal repair performed concurrently with ACLr has an increased durability than those performed without concurrent ACLr. The goal of the current study was to use a large patient database to estimate the failure rate for a meniscal repair, as reflected by subsequent knee surgery, and compare these rates of secondary knee surgery between isolated meniscus repairs and meniscus repairs with concurrent ACL reconstructions.

## Methods

### Patient cohorts

The large insurance claims PearlDiver (Colorado Springs, CO, USA) 2007–2017 Mariner database, which captures data from approximately 55 million patients, was utilized. An exemption from the institutional review board was obtained, as the database contains only de-identified patient data.

Study patients were identified by Current Procedural Terminology (CPT) codes for meniscal repair (29882 and 29883) with or without concurrent arthroscopic ACLr (CPT code 29888). Both the isolated meniscus repair and the combined meniscus repair and ACLr cohorts were then queried for secondary surgical procedures of the knee within the following 2 years. Secondary surgeries of the knee included those specifically focused on the meniscus injury (repair, meniscectomy, meniscus transplant) and those that could be utilized to treat the symptomatic patient with meniscus deficiency (high tibial osteotomy, unicompartmental knee arthroplasty, and total knee arthroplasty) [Table 1].

**Table 1. Description of study population.**

| | |
|---|---|
| **Meniscus Repair with ACL Reconstruction Procedure Codes** | |
| **(CPT-29882 or CPT-29883) AND CPT-29888** | |
| CPT-29882 | Arthroscopy knee surgical; with meniscus repair (medial OR lateral) |
| CPT-29883 | Arthroscopy knee surgical; with meniscus repair (medial AND lateral) |
| CPT-29888 | Arthroscopically aided anterior cruciate ligament repair/augmentation or reconstruction |
| **Meniscus Repair without ACL Reconstruction Procedure Codes** | |
| **(CPT-29882 or CPT-29883) NOT CPT-29888** | |
| CPT-29882 | Arthroscopy knee surgical; with meniscus repair (medial OR lateral) |
| CPT-29883 | Arthroscopy knee surgical; with meniscus repair (medial AND lateral) |
| **Description of Secondary Knee Surgery Procedure Codes** | |
| **Any of the following** | |
| CPT-27440 | Arthroplasty knee tibial plateau; |
| CPT-27441 | Arthroplasty knee tibial plateau; with debridement and partial synovectomy |
| CPT-27442 | Arthroplasty femoral condyles or tibial plateau(s) knee; |
| CPT-27443 | Arthroplasty femoral condyles or tibial plateau(s) knee; with debridement and partial synovectomy |
| CPT-27445 | Arthroplasty knee hinge prosthesis (eg. Walldius type) |
| CPT-27446 | Arthroplasty knee condyle and plateau; medial OR lateral compartment |
| CPT-27447 | Arthroplasty knee condyle and plateau; medial AND lateral compartments with or without patella resurfacing (total knee arthroplasty) |
| CPT-27457 | Osteotomy proximal tibia including fibular excision or osteotomy (includes correction of genu varus (bowleg) or genu valgus (knock-knee)); after epiphyseal closure |
| CPT-29868 | Arthroscopy knee surgical; meniscal transplantation, (medial OR lateral) |
| CPT-29880 | Arthroscopy knee surgical; with meniscectomy (medial AND lateral including any meniscal shaving) |
| CPT-29881 | Arthroscopy knee surgical; with meniscectomy (medial OR lateral including any meniscal shaving) |
| CPT-29882 | Arthroscopy knee surgical; with meniscus repair (medial OR lateral) |
| CPT-29883 | Arthroscopy knee surgical; with meniscus repair (medial AND lateral) |

## Data analysis

Data was analyzed in two ways: all patients identified and those who were confirmed to have remained active in the database for a minimum of two years after the index procedure. Confirmed activity was indicated by the patient being seen for any medical condition at or after two years and therefore guarantees two-year survivorship with or without a secondary procedure.

Frequency, age distribution, rates of secondary surgery, and type of secondary procedures performed were assessed and compared using Fisher exact tests and t-tests, as appropriate. Finally, Kaplan Meier Survival curves were created for the larger cohort and the cohort with 2

years of minimum follow up to evaluate isolated meniscus repair and meniscus repair with combined ACL reconstruction. Comparison of the two groups for each of the sets of cohorts was performed with Log-Rank (Mantel-Cox) test.

Stata v.14 (Stata Corporation, College Station, Texas) was used for analysis. Significance was defined with a two-sided alpha level of $\leq 0.05$.

## Results

### Study cohorts

In total, 1,585 patients undergoing meniscal repair were identified (Fig 1). Isolated meniscal repair was performed for 579 (36.5%) and meniscal repair with ACLr was performed for 1,006 (63.5%). Average follow up was 13 years in the overall cohort, with a range of 1.5 to 24 years. Minimum of two year follow up was present for 487 (30.7% of the overall study population). There was no significant difference in the frequency of procedure performed between the larger cohort and sub-cohort (p = 0.8756).

Age distribution of the combined meniscus repair and ACL reconstruction and isolated meniscus repair are shown in Fig 2 for both the larger cohort (Fig 2A), and the cohort with 2 year follow up (Fig 2B). The majority of patients were between 15 and 24 years old (50.4%,

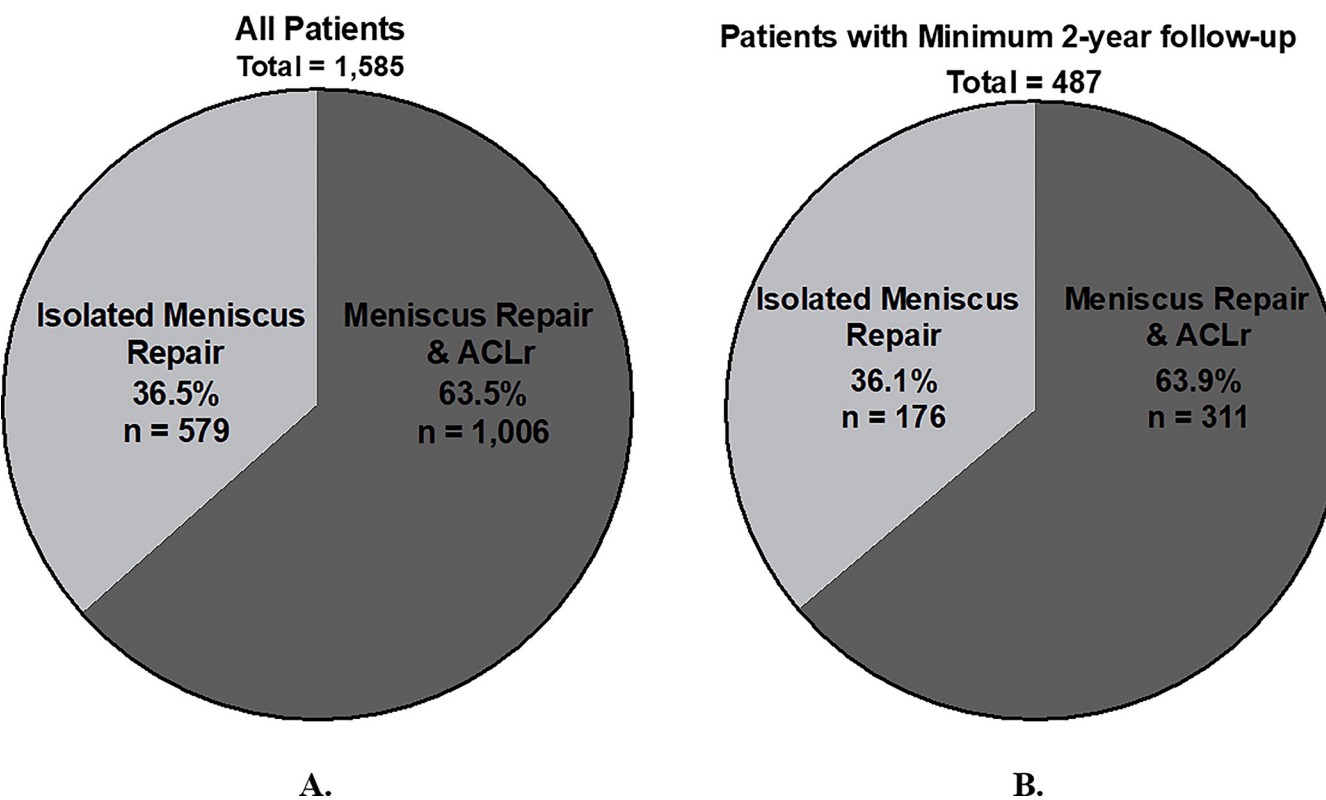

**Fig 1. Comparison of study population, patients who underwent isolated meniscus repair versus meniscus repair with ACL reconstruction.** Comparison of patients who underwent isolated meniscus repair versus meniscus repair with ACL reconstruction is shown for all patients that met inclusion criteria based on procedural codes and age (A), and that same population after filtering for patients with a minimum of 2-years follow-up (B). There were no significant differences in the frequency of procedure performed between the larger cohort and sub-cohort (p = 0.8756).

## Age Distribution of Patients Undergoing Meniscal Repair

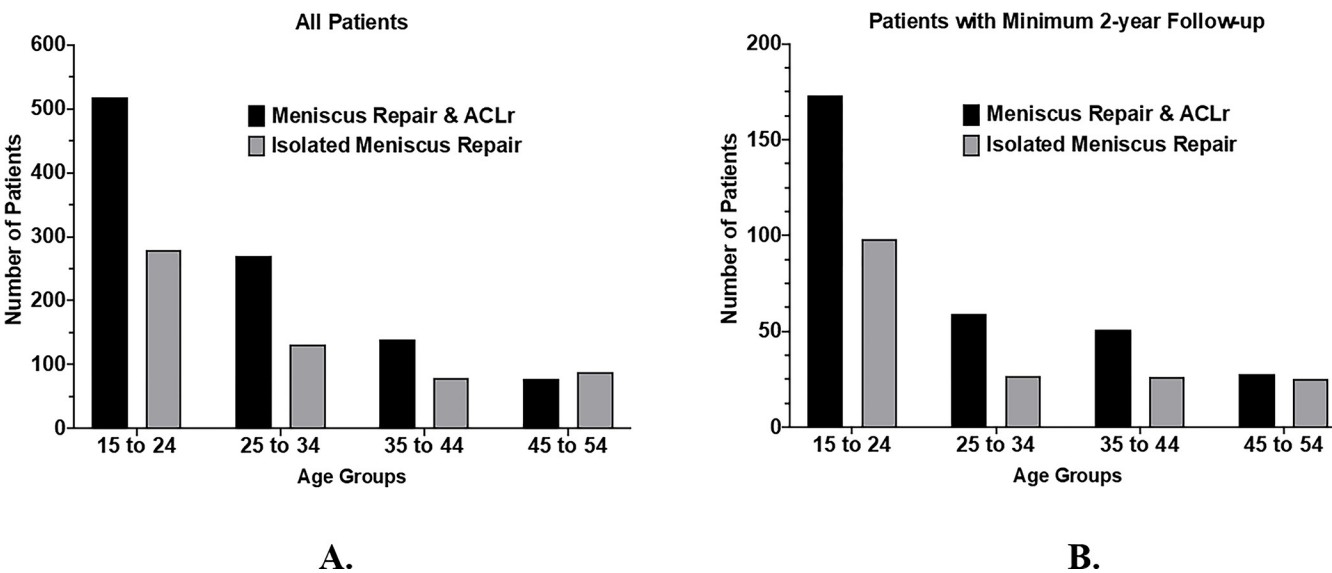

**Fig 2. Age distribution of patients undergoing meniscus repair.** Comparison of age distributions who patients who underwent isolated meniscus repair to those who underwent meniscus repair with ACL reconstruction (all patients [A], and those with minimum 2 year follow up [B]).

n = 799). In the larger cohort, a significantly higher proportion of patients in the older age group (45–54) underwent isolated meniscus repair (15.20% vs. 7.75%, p < 0.0001); otherwise, there were no other differences in age distribution. This trend is also present in the smaller study cohort with 2 year follow up (14.20% vs. 9.00%), although the difference was not statistically significant (p = 0.0951).

### Secondary surgeries

Rates of secondary surgeries were then assessed (**Fig 3**). For the larger cohort, secondary surgeries were performed for 110/1,006 (10.9%) of those with meniscal repairs with concurrent ACLr, as compared to 79/579 (13.6%) of those with isolated meniscal repair (p = 0.126). For the 2 year follow up cohort, secondary surgery rates were also not significantly different (19.3% vs. 25.6%, p = 0.1098).

Of those revision surgeries, approximately 80% were meniscectomies across all patients and cohorts (**Table 2**). There were no differences in type of revision surgery that could be detected between any of the analyzed cohorts. No meniscus transplants or high tibial osteotomies were subsequently needed and between 1–9 patients had a total knee arthroplasty. Because of privacy compliance in the PearlDiver database, any category of patients with less than 10 are not fully available, which is why a more precise reporting of those secondary surgeries is not available for further analysis, nor for sub-analyses based on patient demographics.

Kaplan Meier Survivor curves are shown in **Fig 4**. There were no differences in the pattern of survivorship patterns between the two procedures in either cohort set (p = 0.2016 and p = 0.0586).

### Discussion

The importance of maintaining the meniscus when possible is well documented [5–8,10,20,21] Meniscal repair, as opposed to debridement, is thus frequently considered [22–25]. Not only

## Secondary Surgeries After Meniscal Repair

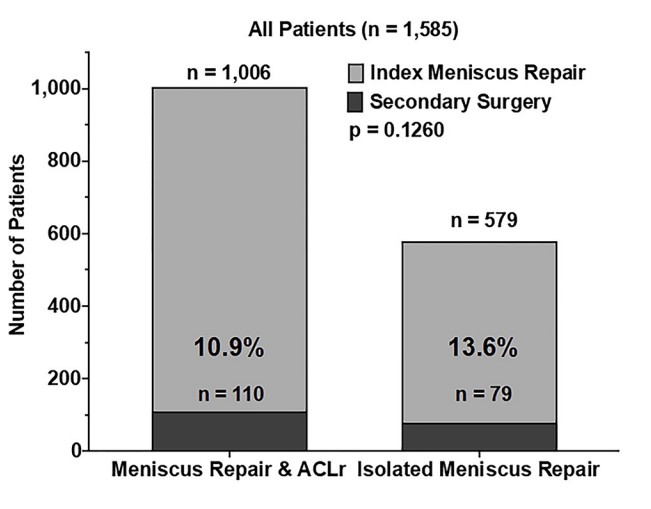

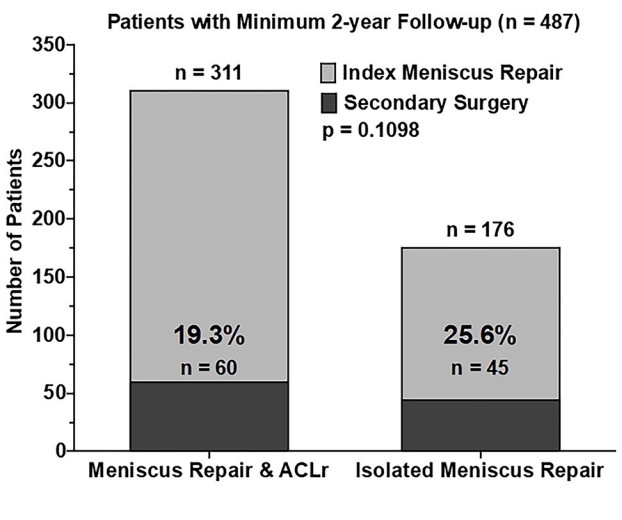

**A.**

**B.**

**Fig 3. Secondary surgeries after meniscal repair.** Secondary surgery rates in patients who underwent isolated meniscus repair versus meniscus repair with ACL reconstruction. Results are shown for all patients (A), and those with a minimum of 2-years follow-up (B). Secondary surgery rate was not significantly different between those undergoing meniscal repair with or without ACL reconstruction (all patients p = 0.1260, those with minimum two year follow up p = 0.1098).

does this scenario become frequently considered in the presence of ACL injury, some [10,18] but not all [9,19–21] studies have suggested that the meniscal repair healing may be improved in the biologic milieu of concurrent ACLr. In the current large database study, we found no significant difference in rates of secondary knee surgery between meniscus repairs performed in isolation and those repaired with combined ACL reconstruction.

In the evaluated dataset, approximately two-thirds of meniscal repairs were performed concurrent with ACLr. Data was analyzed for all patients in the database regardless of follow up, in addition to patients with a minimum of two year follow up, and no differences were found between the larger and smaller cohorts.

There was a predilection of meniscal repairs in conjunction with ACLr in the younger patients. Historically, several studies have reported less favorable results with meniscal repairs in the older population [26,27]. Additionally, older adults with ACL ruptures are more often treated conservatively [28], without surgical intervention, and so the current study's findings of meniscal repair with concurrent ACLr in younger patients is not surprising.

The rate of secondary surgeries after meniscal repairs were not found to be significantly different between those with or without concurrent ACLr. The average rate of secondary surgeries for the entire cohort was 11.9% and for those with a minimum of two year follow up was 21.6%. These numbers are in line with existing literature [9,10,18–21].

The current study is in line with several studies that have not found differences in secondary surgery rates between meniscal repairs with or without concurrent ACLr [9,19–21]. Bogunovic et al. investigated outcomes of 75 meniscal repairs using an all-inside technique in both isolated meniscus repair and combined meniscus repairs and ACL reconstructions [19]. They found that 84% of patients remained asymptomatic and that there were no significant differences in failures between meniscus repairs with or without concurrent ACL reconstruction [19].

**Table 2. Breakdown of secondary procedures.**

| Code | Description | All Patients (1,006): n (%) | Patients >2 Years Follow-up (311): n (%) |
|---|---|---|---|
| **Breakdown of Secondary Procedures after Meniscus Repair & ACLr** | | | |
| | **Total** | **110 (10.9)** | **60 (19.3)** |
| CPT-29881 | Arthroscopy knee surgical; with meniscectomy (medial OR lateral including any meniscal shaving) | 75 (68.2) | 48 (80.0) |
| CPT-29882 | Arthroscopy knee surgical; with meniscus repair (medial OR lateral) | 27 (24.6) | <10 (6.7) |
| CPT-29880 | Arthroscopy knee surgical; with meniscectomy (medial AND lateral including any meniscal shaving) | 11 (10.0) | <10 (6.7) |
| CPT-29883 | Arthroscopy knee surgical; with meniscus repair (medial AND lateral) | <10 (1.0) | <10 (6.7) |
| CPT-27447 | Arthroplasty knee condyle and plateau; medial AND lateral compartments with or without patella resurfacing (total knee arthroplasty) | 0 (0) | 0 (0) |

| Code | Description | All Patients (579): n (%) | Patients >2 Years Follow-up (176): n (%) |
|---|---|---|---|
| **Breakdown of Secondary Procedures after Isolated Meniscus Repair** | | | |
| | **Total** | **79 (13.6)** | **45 (25.6)** |
| CPT-29881 | Arthroscopy knee surgical; with meniscectomy (medial OR lateral including any meniscal shaving) | 65 (82.3) | 39 (86.7) |
| CPT-29882 | Arthroscopy knee surgical; with meniscus repair (medial OR lateral) | 12 (15.2) | <10 (4.4) |
| CPT-27447 | Arthroplasty knee condyle and plateau; medial AND lateral compartments with or without patella resurfacing (total knee arthroplasty) | <10 (1.25) | <10 (4.4) |
| CPT-29880 | Arthroscopy knee surgical; with meniscectomy (medial AND lateral including any meniscal shaving) | <10 (1.25) | <10 (4.4) |
| CPT-29883 | Arthroscopy knee surgical; with meniscus repair (medial AND lateral) | 0 (0) | 0 (0) |

Note: Some patients underwent more than one secondary procedure.

In contrast, the current study is in distinction to other studies that have found differences in secondary surgery rates between meniscal repairs with or without concurrent ACLr [10,18]. Wasserstein et al., in a matched cohort study of approximately 2500 total patients, found higher rates of secondary knee surgery in patients undergoing isolated meniscus repair, compared to combined meniscus repair and ACL reconstruction at 2 year follow up (16.7% vs. 9.7%, p < 0.0001) [10]. Similarly, Ronnblad et al. noted less failure of meniscal repair when simultaneous ACL reconstruction was performed (7% absolute and 42% relative risk reduction of reoperation after 2 years compared with isolated meniscal repair) [29]. However, the present study did note a nonsignificant trend towards less subsequent surgery rates in patients with concurrent ACLr, similar to the studies reviewed.

Despite a lack of published evidence, some authors postulate that the higher healing rates of meniscus repairs with combined ACL reconstruction are due to the rich biologic environment created during reconstruction [10,17,18]. This theory has led to additional techniques, such as notch microfracture with isolated meniscal repair as a means to augment healing via bone marrow stimulation [30]. Fibrin clot, created by spinning autologous blood in a tube until a clot is formed, has also been proposed as an adjunct to healing. Henning et al showed that incorporation of a fibrin clot into an isolated meniscal repair resulted in a failure rate of 8%, compared with 41% without the clot [31]. Other adjuncts such as platelet-rich plasma (PRP)

## Survivorship Curves of Index Meniscal Repairs

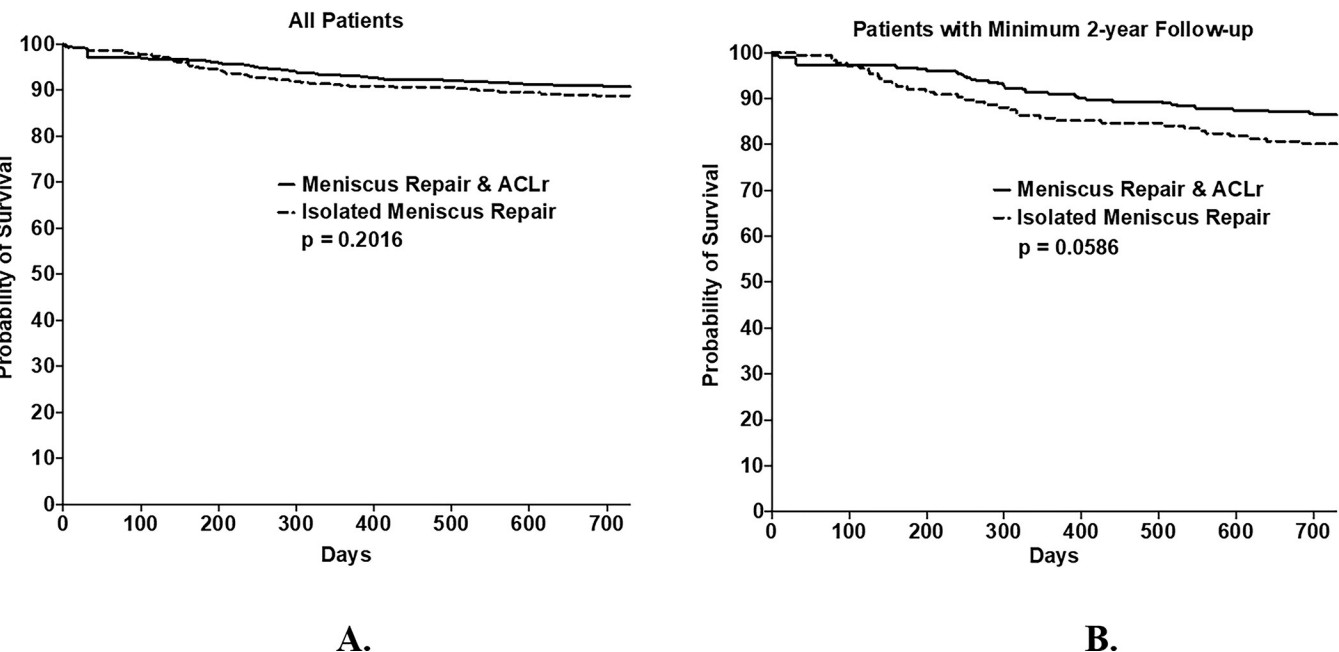

**A.** **B.**

**Fig 4. Survivorship curves of index meniscal repairs.** Kaplan Meier Survival curve for those undergoing meniscal repair with or without ACL reconstruction for all patients (A) and those with a minimum of two years of follow up (B). Rate of secondary surgery at two years is not significantly different between group (all patients p = 0.2016, those with minimum two years of follow up p = 0.0586).

have shown no improvement in self-reported knee function or objective functional testing in patients with combined ACLr and meniscus repair [32]. However, Everhart et al. found that PRP had a protective effect against isolated meniscal repair failure but similarly found no significant benefit when combined with ACLr [33].

When assessing all meniscus repairs in this large database study, we found a 11.9% rate of secondary surgery. For those patients who were followed for a minimum of 2 years, we demonstrated an overall 21.6% rate of secondary knee surgery. Overall, this 21.6% rate of secondary knee surgery at 2 year follow up is consistent with prior literature. Everhart et al., in a study of 235 meniscus repairs treated by a single surgeon, demonstrated a 20.2% failure rate at 5 year follow up [21]. Of those 235 patients, 73% underwent combined meniscus repair and ACL reconstruction, which is slightly higher than the frequency of combined procedures in the current study. Additionally, Everhart et al. demonstrated no difference in failure rates between combined meniscus repairs and ACL reconstructions, and isolated meniscus repairs [21]. In a meta-analysis of meniscus repairs with minimum of 5-year follow, Nepple et al demonstrated a 22.3% to 24.3% failure rate of 566 patients sampled [20]. However, they failed to find a statistically significant difference in failure between combined meniscus repair and combined ACL reconstruction (26.9%) compared to isolated meniscus repairs (22.7%) [20].

Analysis of Kaplan Meier Survival curves demonstrated that there was no significant difference in pattern of failure between combined meniscus repair and combined ACL reconstruction and isolated meniscus repairs. Both cohorts demonstrated a gradual failure over time and not a significant decline at any specific time period.

The limitations of this study are inherent to a database study. The database is only able to capture insured patients, though with both public and private payers and over 55 million lives

covered nationally, the large size augments external validity. Additionally, patients that changed insurance status or did not follow up may be lost within the 2 year follow up time period of this study, thus the secondary analysis was performed on those with a minimum 2 year follow up. Furthermore, the inclusion of patients in this study was reliant on the accuracy of the CPT coding, and surgical or clinical data were unavailable, as with most large database studies. This study is unable to stratify the meniscus tear pattern that was repaired, the type of fixation device used to perform the repair as they are indistinguishable based on the CPT code. As the query of CPT codes does not allow for modifiers, laterality data was not available. Further, the evaluation of subsequent surgery rates may be underestimated, as this study purposely did not include subsequent surgeries for microfracture, osteochondral grafting, chondroplasty, or revision ACL repair, as these CPT codes created too much noise when evaluated, and decision was made to only include surgeries directly related to the meniscus and definitive endpoints such as total knee replacements. Thus the subsequent surgery rates in this study may be underestimated.

In conclusion, using a large database this study found that meniscus repairs fail at a rate of 21.6% at 2 year follow up, based on the occurrence of secondary knee surgery. Additionally, based on this data, there is no difference between failure rates of meniscus repairs performed alone or with a concurrent ACLr.

## Author Contributions

**Conceptualization:** Joseph B. Kahan, Patrick Burroughs, Christopher A. Schneble, William Mclaughlin, Adam Nasreddine, Jonathan N. Grauer, Michael J. Medvecky.

**Data curation:** Patrick Burroughs, Logan Petit, Jay Moran.

**Formal analysis:** Peter Joo, Jay Moran.

**Investigation:** Joseph B. Kahan, Logan Petit, Christopher A. Schneble, Michael J. Medvecky.

**Methodology:** Joseph B. Kahan, Patrick Burroughs, Logan Petit, Christopher A. Schneble, Maxwell Modrak, Adam Nasreddine, Jonathan N. Grauer, Michael J. Medvecky.

**Project administration:** Peter Joo.

**Resources:** Patrick Burroughs, Maxwell Modrak.

**Software:** Jay Moran.

**Supervision:** William Mclaughlin, Jonathan N. Grauer, Michael J. Medvecky.

**Validation:** Jay Moran, William Mclaughlin.

**Visualization:** Jay Moran.

**Writing – original draft:** Joseph B. Kahan, Patrick Burroughs, Logan Petit, Christopher A. Schneble, Peter Joo, Jay Moran, Maxwell Modrak, William Mclaughlin, Adam Nasreddine, Jonathan N. Grauer, Michael J. Medvecky.

**Writing – review & editing:** Joseph B. Kahan, Logan Petit, Peter Joo, Jay Moran, Maxwell Modrak, William Mclaughlin, Adam Nasreddine, Michael J. Medvecky.

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
