## [Decision Letter · Decision Letter 0]

28 Jul 2023

PONE-D-23-15574Subsequent Surgical Rates of Meniscus Repair with and without Anterior Cruciate Ligament ReconstructionPLOS ONE

Dear Dr. Medvecky,

Thank you for submitting your manuscript to PLOS ONE. After careful consideration, we feel that it has merit but does not fully meet PLOS ONE’s publication criteria as it currently stands. Therefore, we invite you to submit a revised version of the manuscript that addresses the points raised during the review process.

We look forward to receiving your revised manuscript.

Kind regards,

Hassan Zmerly, MD PhD

Academic Editor

PLOS ONE

Journal Requirements:

“I have read the journal's policy and the authors of this manuscript have the following competing interests: Medvecky guest speaker/honoraria for Smith & Nephew”

Reviewers' comments:

Reviewer's Responses to Questions

**Comments to the Author**

1. Is the manuscript technically sound, and do the data support the conclusions?

Reviewer #1: Yes

Reviewer #2: No

2. Has the statistical analysis been performed appropriately and rigorously? 

Reviewer #1: Yes

Reviewer #2: Yes

3. Have the authors made all data underlying the findings in their manuscript fully available?

Reviewer #1: Yes

Reviewer #2: No

4. Is the manuscript presented in an intelligible fashion and written in standard English?

Reviewer #1: Yes

Reviewer #2: No

5. Review Comments to the Author

Reviewer #1: Thank you for a study with a pertinent question, large database, and a sound discussion.

Questions:

1. For secondary knee surgery procedure codes used to query for second surgeries within 2 years, did you consider looking for cartilage-injury treatment codes, like chondroplasty, microfracture, osteochondral grafting (autograft and allograft)? Those can be considered to be associated with meniscal repair failures along with the secondary procedures you chose to query.

Also, revision ACL reconstruction? It seems unusual that among the sub-cohort of 1006 patients who underwent ACL reconstruction there were no re-tears for whom a revision ACL reconstruction was performed.

If you decided against querying for those codes, would be interested in your reasoning.

2. Would there be a way to add a figure and some discussion on age-specific rates of second surgeries (and types)? There might be some interesting findings there.

Since you did a nice job showing age distribution of all patients and those with 2 year follow up in Figure 2, I thought it would be possible to also show results distribution by age, and was hoping to see that.

Thank you!

Reviewer #2: I would first thank the author for this manuscript, however few comments need to be considered

1- The title of the article is misleading and needs to be reconsidered, I understood the purpose of the study after I read the abstract. eg, The need for subsequent surgery after arthroscopic meniscal repair with or without ACL reconstruction OR The failure rates after meniscal repair surgery with or without ACL reconstruction.

2- Linguistic review is needed

3- The author should avoid abbreviations like ACLr and needs to clarify it in abstract.

4- Line 55 - 75 need to be clarified. How many intervals did the author reported the outcome.

5- Significant drop of the study population, why is that?

6- Table 1 is not really adding to the paper and instead the authors could have mentioned the techniques used for meniscal repair in their population.

7- I understood that you measured the outcomes in two cohorts, while the second cohort is understood to be at 2 years, what was the follow up interval of the larger cohort?

8- Line 179 - Between 1-9 patient had TKA. this is weak point of the study. why you are not precise?

6. PLOS authors have the option to publish the peer review history of their article (what does this mean?). If published, this will include your full peer review and any attached files.

Reviewer #1: No

Reviewer #2: **Yes: **Aissam Elmhiregh

---

## [Author Response · Author response to Decision Letter 0]

6 Sep 2023

Ref.: Ms. No. PONE-D-23-15574

Journal: PLOS ONE

Title: Rates of subsequent surgeries after meniscus repair with and without anterior cruciate ligament reconstruction

Dear Reviewers,

We thank the PLOS ONE reviewers for their insightful comments, which have helped to make the manuscript better. We have included an itemized list with our responses to all the reviewers’ comments on how we have altered the manuscript text to address these concerns.

Reviewer Responses

Reviewer #1 

Thank you for a study with a pertinent question, large database, and a sound discussion.

Questions:

1. For secondary knee surgery procedure codes used to query for second surgeries within 2 years, did you consider looking for cartilage-injury treatment codes, like chondroplasty, microfracture, osteochondral grafting (autograft and allograft)? Those can be considered to be associated with meniscal repair failures along with the secondary procedures you chose to query. Thank you for this comment. We had considered adding cartilage-injury treatment codes as secondary surgeries, but due to the increase in noise in the dataset from addition of each additional CPT code, the research team made a decision to focus secondary procedures to those directly related to the index procedure (repeat meniscal surgery) or definite end point (arthroplasty) as this would provide the cleanest results from this particular dataset.

Also, revision ACL reconstruction? It seems unusual that among the sub-cohort of 1006 patients who underwent ACL reconstruction there were no re-tears for whom a revision ACL reconstruction was performed.

If you decided against querying for those codes, would be interested in your reasoning. This is absolutely a limitation of this study, and a statement has been added to the discussion section expanding on this point. Revision ACL reconstruction is a valuable point to consider, but when the CPT code was added, an error occurred for the ACLr cohort as the initial inclusion criteria query had only pulled ACLr reports for the index procedure.

This consideration that our subsequent revision rates may be an underestimate and does not apply to patients undergoing revision ACL reconstruction has been clarified.

2. Would there be a way to add a figure and some discussion on age-specific rates of second surgeries (and types)? There might be some interesting findings there.

Since you did a nice job showing age distribution of all patients and those with 2 year follow up in Figure 2, I thought it would be possible to also show results distribution by age, and was hoping to see that.

Thank you! We agree this is an interesting point and had attempted to subcategorize the secondary surgeries, but given the HIPAA restrictions on reporting database categorization data with <10 patients, the precise reporting of age-specific rates was not available for secondary analysis as described in the manuscript.

Reviewer #2 

I would first thank the author for this manuscript, however few comments need to be considered

1- The title of the article is misleading and needs to be reconsidered, I understood the purpose of the study after I read the abstract. eg, The need for subsequent surgery after arthroscopic meniscal repair with or without ACL reconstruction OR The failure rates after meniscal repair surgery with or without ACL reconstruction. Thank you for this comment. The title of the manuscript has been revised to: Rates of subsequent surgeries after meniscus repair with and without concurrent anterior cruciate ligament reconstruction. This better clarifies the purpose of this study.

2- Linguistic review is needed We appreciate this comment and have combed through the manuscript for linguistic review

3- The author should avoid abbreviations like ACLr and needs to clarify it in abstract. The abbreviation has been clarified in the abstract

4- Line 55 - 75 need to be clarified. How many intervals did the author reported the outcome. Thank you, the mean and range of overall follow up has been included

5- Significant drop of the study population, why is that? As described in the methods and limitations, performing statistics with a large database requires stringent inclusion criteria. One such inclusion criteria was to ensure at least 2 year follow up, which will knowingly reduce the study population if their surgery was within 2 years, changed insurance, or were lost to follow up within 2 years. While this purposefully reduces the study population, the statistics performed within this sub-population is much more accurate.

6- Table 1 is not really adding to the paper and instead the authors could have mentioned the techniques used for meniscal repair in their population. We appreciate the reviewer’s comment. However, given this is a database study, the primary method of identifying meniscal repairs is via CPT codes. Unfortunately this means that the techniques used by surgeons is unavailable, as this is a national administrative database analysis and not a chart review study

7- I understood that you measured the outcomes in two cohorts, while the second cohort is understood to be at 2 years, what was the follow up interval of the larger cohort? Thank you for this point – we have included the follow up interval within the abstract and the results section for the overall cohort as a mean of 13 years (1.5-24 years)

8- Line 179 - Between 1-9 patient had TKA. this is weak point of the study. why you are not precise? Thank you for this comment. As is described in the next sentence, national policy on privacy (HIPAA) prevents reporting of database study results of <10 patients, as there is potential to be deemed identifiable. Thus database studies by convention commonly report “less than 10” for such results.

---

## [Decision Letter · Decision Letter 1]

13 Nov 2023

Rates of subsequent surgeries after meniscus repair with and without concurrent anterior cruciate ligament reconstructionDear Dr. Michael MedveckyWe are pleased to inform you that your manuscript has been judged scientifically suitable for publication and will be formally accepted for publication once it meets all outstanding technical requirements.Within one week, you'll receive an e-mail detailing the required amendments. When these have been addressed, you'll receive a formal acceptance letter and your manuscript will be scheduled for publication.An invoice for payment will follow shortly after the formal acceptance. To ensure an efficient process, please log into Editorial Manager at http://www.editorialmanager.com/pone/, click the Update My Information link at the top of the page, and double check that your user information is up-to-date. If you have any billing related questions, please contact our Author Billing department directly at authorbilling@plos.org.If your institution or institutions have a press office, please notify them about your upcoming paper to help maximize its impact. If they’ll be preparing press materials, please inform our press team as soon as possible -- no later than 48 hours after receiving the formal acceptance. Your manuscript will remain under strict press embargo until 2 pm Eastern Time on the date of publication. For more information, please contact onepress@plos.org.Kind regards,Hassan Zmerly, MD PhDAcademic Editor, PLOS ONE

---

## [Editor Report · Acceptance letter]

16 Nov 2023

PONE-D-23-15574R1 

Rates of subsequent surgeries after meniscus repair
with and without concurrent anterior cruciate ligament reconstruction 

Dear Dr. Medvecky:

I'm pleased to inform you that your manuscript has been deemed suitable for publication in PLOS ONE. Congratulations! Your manuscript is now with our production department. 

Kind regards, 

on behalf of

Professor Hassan Zmerly 

Academic Editor

PLOS ONE